# Incorporation of Organic Growth Additives to Enhance In Vitro Tissue Culture for Producing Genetically Stable Plants

**DOI:** 10.3390/plants11223087

**Published:** 2022-11-14

**Authors:** Imtinene Hamdeni, Mounir Louhaichi, Slim Slim, Abdennacer Boulila, Taoufik Bettaieb

**Affiliations:** 1Research Laboratory of Horticultural Sciences, National Agronomic Institute of Tunisia, University of Carthage, Tunis 1082, Tunisia; 2International Center for Agricultural Research in the Dry Areas (ICARDA), Tunis 1004, Tunisia; 3Department of Animal and Rangeland Science, Oregon State University, Corvallis, OR 97331, USA; 4Research Unit of Biodiversity and Valorization of Resources in Mountainous Areas, School of Higher Education in Agriculture of Mateur, University of Carthage, Mateur 7030, Tunisia; 5Laboratoire des Substances Naturelles, Institut National de Recherche et d’Analyse Physico-Chimique, Biotechpole de Sidi Thabet, Ariana 2020, Tunisia

**Keywords:** in vitro, plant propagation, afforestation, organic growth additives, genetic stability, molecular markers, large scale restoration

## Abstract

The growing demand for native planting material in ecological restoration and rehabilitation for agro-silvo-pastoral ecosystems has resulted in a major global industry in their sourcing, multiplication, and sale. Plant tissue culture is used for producing high-quality, disease-free, and true-to-type plants at a fast rate. Micropropagation can help to meet the increasing demand for planting material and afforestation programs. However, in vitro plant propagation is an expensive technique compared to conventional methods using suckers, seeds, and cuttings. Therefore, adopting measures to lower production costs without compromising plant quality is essential. This can be achieved by improving the culture media composition. Incorporating organic growth additives can stimulate tissue growth and increase the number of shoots, leaves, and roots in culture media. Organic growth supplementation speeds up the formation and development of cultures and yields vigorous plants. Plant regeneration from meristems (shoot tips and axillary buds) is a reliable way to produce true-to-type plants compared with callus and somatic embryogenesis regeneration, but in vitro culture environments can be mutagenic. Therefore, detecting somaclonal variations at an early stage of development is considered crucial in propagating plants. The genetic stability of in vitro regenerated plants needs to be ascertained by using DNA-based molecular markers. This review aims to provide up-to-date research progress on incorporating organic growth additives to enhance in vitro tissue culture protocols and to emphasize the importance of using PCR-based molecular markers such as RAPD, ISSR, SSR, and SCoT. The review was assessed based on the peer-reviewed works published in scientific databases including Science Direct, Scopus, Springer, JSTOR, onlinelibrary, and Google Scholar.

## 1. Introduction

Climate change and induced human activities have negatively impacted agro-silvo-pastoral ecosystems across the planet [1,2]. The global push to achieve ecosystem restoration targets has resulted in an increased demand for native plant material that current production systems are unable to satisfy [3]. Using native species in restoration efforts is critical for recreating or maintaining healthy, resistant, and resilient ecosystems and communities [4]. Therefore, the development of methods for the large-scale plantations of selected genotypes of medicinal, pastoral, and forest species has become increasingly important in view of the need for the rehabilitation of marginal and degraded rangelands [5,6,7]. Regeneration of plants via in vitro tissue culture is considered to be an efficient approach for clonal plant propagation. Developing protocols for successful plant tissue culture is complicated because there are various interacting factors. Plant raw material, culture conditions, and culture media composition are determining factors in the quality of the final product obtained in any plant cell culture protocol [8]. The choice of nutritional components and growth regulators is one of the most important factors governing the growth and morphogenesis of the plant tissues in culture [9]. There are many components and additives used in plant micropropagation media which vary according to the plant species, cultivar, or explant type and must be experimentally defined for each case [10]. For the optimal growth of tissues in vitro, nutritional requirements should be present in optimum concentrations [11]. Micropropagation media are generally made up of these components: macronutrients, micronutrients, vitamins, amino acids, sugar, gelling agents, and growth regulators [12]. However, these compounds are expensive and pose risks if added in inadequate amounts [13]. Therefore, many studies have been conducted to explore the modification of culture media composition by adding low-cost organic materials as an alternative to expensive materials without compromising the quality of the produced plants [14,15,16]. A variety of organic growth additives such as coconut water, banana pulp, yeast extract, tomato juice, papaya juice, potato homogenate, and pineapple pulp can be very effective in promoting plant growth and development [17,18]. Such organic growth additives provide undefined mixtures of organic nutrients and growth factors [19]. The reasons for applying organic growth additives to the culture medium, besides being a natural source of carbon, are because they contain natural vitamins, phenols, fiber, hormones, proteins, lipids, and minerals [10]. An overview of culture media composition modification for plant tissue propagation with special regard to organic growth additives and the importance of using PCR-based molecular markers to establish the genetic stability of in vitro regenerated plants are discussed in this paper. The literature was obtained through a keyword-based search on Science Direct, Scopus, Springer, JSTOR, onlinelibrary, and Google Scholar. All databases up to July 2022 are considered.

## 2. Supplementation of Organic Growth Additives to Enhance In Vitro Culture Techniques

Adding organic growth additives to culture media, individually or in combination, accelerates shoot initiation and proliferation. Depending on the species and genotype, different types and concentrations of organic nutrients are needed for the success of in vitro cultures (Table 1) [12,20,21,22]. Several studies were conducted to evaluate the nutritional role of organic additives and their impact on shoot regeneration and growth to induce rhizogenesis and produce viable plantlets [17,23].

### 2.1. Vegetable, Fruit, and Plant Extracts 

The growth and development of tissues vary for different plants according to their nutritional requirements. Tissues from different parts of plants may also have different requirements for satisfactory growth. Commonly, culture media are supplemented with a variety of organic substances or extracts including coconut milk, pineapple pulp, papaya juice, banana homogenate, orange juice, and tomato juice, to test their effect on growth enhancement [49].

Wiszniewska et al. [24] reported the successful micropropagation protocol of three *Daphne* species (*Daphne caucasica*, *Daphne tangutica*, and *Daphne jasminea*) on MS media supplemented with organic growth additives, including coconut water and pineapple pulp. Daud et al. [25], on in vitro regeneration of *Celosia* sp. using five kinds of organic growth additives, showed that young coconut juice (YCJ) at 70 mL/L induced the highest shoot regeneration (14.21), banana and tomato juices promoted the highest shoot regeneration of stem segments at 50 mL/L which produced 9.57 and 9.28 shoots per explants, and papaya juice, at the lowest concentration (20 mL/L), showed the highest shoot regeneration (10.5). Different organic growth additives such as coconut water, coconut milk, grind spinach leaves, grind potato tubers, grind carrot, rice flour, green gram, grind pumpkin, banana fruit, and orange juice were assayed by Manawadu et al. [14] to enhance in vitro regeneration of *Raphanus sativus*. The best response was observed in MS supplemented with 10% orange juice which produced the highest number of shoots (12 shoots/explant). Banana powder (BW), coconut water (CW), and potato dextrose (PD) were added to a basal seed sowing media of *Epidendrum nocturnum*. The culture medium with 10 mL/L CW showed the greatest germination percentage (71% and 76.75%) compared to 60 and 90 days after seed sowing. Media with 5 g/L BW + 5 mL/L CW showed greater values of plant length (19.80 mm), number of roots (2.1), and fresh weight (0.08 g) [16]. The effect of various organic additives (coconut water, birch sap, maple sap, and banana powder) on in vitro germination, protocorm formation, and seedling growth from *Cypripedium macranthos* on one-quarter MS medium was investigated during asymbiotic seed culture. The highest germination and protocorm formation percentages were achieved (70.8% and 74.2%) when 100 mL/L of coconut water was added to the basal medium. With 100 mL/L of birch sap or maple sap, the germination and protocorm formation percentages were over 65% and 68% [15]. Knudson C media supplemented with organic growth additives (coconut water, tomato juice, and banana pulp) at different concentrations for *Dendrobium lowii* in vitro propagation showed a notable protocorm development when treated with 25 g/L of banana pulp compared to the other supplements. It showed the highest growth index value of 593.3 with 100% of protocorms that successfully developed shoots and 93.3% of protocorms producing roots [26].

Supplementing coconut water (50 and 100 mL/L) for *Musa* cv. Rajabulu in vitro propagation showed the best results on the number of roots (9.33; 9) and root length (11.6; 10.76 cm). With CW concentrations from 50 to 200 mL/L, acclimatization was 100% [27]. Selakorn et al. [18] found that when CW (at 200 mL/L) was supplemented to MS medium, the highest number of shoots/explant (1.71), the highest shoot length (4.25 cm), and the highest number of leaves/explant (4) were recorded on *Musa acuminata* in vitro multiplication. For mass propagation of the endangered orchid *Dendrobium chryseum*, supplementing 10% CW to one-half MS medium resulted in the highest number of shoots as well as the longest [28]. The multiplication rate of *Dianthus caryophyllus* increased over four times when the shoot tip and nodal segment were used as explants on MS medium supplemented with 10% CW [29]. However, when 5% CW was added to one-half MS medium, the highest germination percentage for *Gastrochilus matsuran* was 93.3% [30]. Adding 4% coconut water with 3% sucrose increased the shoot elongation of *Hylocereus polyrhizus* to 2.45 cm [31]. MS media enriched with 2% CW enhanced shoot multiplication for *Echinacea purpurea* to 2.58 buds per explant [32].

Adding tomato juice (TJ) to promote shoot regeneration and multiplication of *Physalis angulata* produced the maximum shoot number (12.5) in MS supplemented with 5% TJ while the maximum shoot length (10.7 cm) was obtained with 7.5% TJ [33]. Phytamax media supplemented with 100 mL/L pineapple juice for in vitro propagation of *Laelia rubescens* resulted in the highest seedling height (1.31 cm), the number of leaves per seedling (3.33), and roots (2.33) [34]. However, at 200 mL/L, pineapple juice showed the highest results with 56% asymbiotic germination and 25.8 in seedling formation for in vitro development of *xLaeliocattleya* on MS medium [35].

*Aloe vera* gel (AvG) is the most commonly used part of the plant because of its biological effectiveness and chemical composition (carbohydrates, organic acids, proteins, phenolic compounds, vitamins, minerals, and amino acids) [49]. AvG has gained attention because of its interesting antioxidant and antimicrobial properties. Nowadays, it is proposed as an alternative to conventional additives used in plant micropropagation media [36,37,38]. Hamdeni et al. [36] reported that AvG served as an organic nutritional supplement for the enhancement of *A. vera* in vitro propagation protocol. The work of Haque and Ghosh [37] on *A. vera* showed that the highest number of formed shoots per explant (length ≥ 2 cm) was 17.8 shoots on MS medium supplemented with 10% AvG. For root induction, adding AvG (20% and 30%) to one-third of MS increased the response to 100% rooting, the number of roots per shoot (9.8; 9.2), and the length of the roots (3.1; 2.8). The study on the in vitro micropropagation of *Bacopa chamaedryoides* reported that the best rooting response (100%), number (18.3), and length (2.3 cm) of the shoot were achieved on one-half strength MS medium supplemented with 50% AvG [38].

### 2.2. Amino Acids, Polyamines, and Proteins

Most plants can synthesize the essential requirements of amino acids for cell proliferation and regeneration. Despite this, the exogenous supply of amino acids to culture media plays an important role in stimulating cell growth and the morphogenesis of tissues. Unlike inorganic nitrogen, amino acids are easily assimilated by plant cells and tissues [50]. Saad and Elshahed [51] suggested that plant cells have a higher capacity to take up and transport nitrogen from organic sources rather than inorganic ones. Several studies have reported using amino acids as an organic nitrogen source during in vitro propagation of several species such as *Fragaria × Ananassa* duch cv. Chandler, *Oryza sativa*, *Rosa centifolia*, *Carica papaya*, and *Hibiscus moscheutos* to enhance plant tissue growth and increase their regeneration [52,53,54,55,56]. According to Mandal et al. [39], a combination of all amino acids (methionine, glutamic acid, glycine, tryptophane, proline, lycine, arginine, and glutamine) in 20 mg/L concentrations resulted in the best axillary shoot proliferation response (100%) on *Aegle marmelos* and average shoot number per explant (2.22) when cultured on MS medium supplemented with 2 mg/L BAP. Baskaran et al. [57] reported that the type and amount of amino acids used in the medium have a significant effect on growth and multiple shoot development. Khatri et al. [40] evaluated three nitrogen sources, adenine sulfate (Ads), casein hydrolysate (CH), and putrescine (PU) for their ability to enhance in vitro shoot multiplication of *Chlorophytum borivilianum*. It confirmed that including 20 mg/L of Ads to MS medium supplemented with 2 mg/L BAP + 1 mg/L NAA resulted in the best shoot induction response (96.67%). For rooting, one-half MS supplemented with 9 mg/L PU + 2 mg/L IBA was best with 83.33% root induction. Samiei et al. [41] found that adding 600 mg/L of casein hydrolysate to the Vander Salm medium resulted in the maximum shoot number (4.1 shoots/explant) while glutamic acid at 12 mg/L enhanced shoot regeneration and leaf number of *Rosa canina*. Glutamine at 30 mg/L and asparagine at 20 mg/L concentrations improved shoot multiplication of *Orthosiphon aristatus* on MS medium supplemented with 1 mg/L BAP and 0.5 mg/L Kn [42]. David et al. [43] reported the rapid development from the protocorms of *Vanda helvola* (99.5%) treated in Knudson C medium containing 0.1% peptone, which successfully produced 3.10 leaves with an average length of 10.97 mm per responsive explant after 90 days of culture. Casein hydrolysate at 0.05% was most effective on *Stevia rebaudiana* in vitro propagation, which resulted in 90% regeneration frequency, a maximum of 15 shoots, and a shoot length of 6 cm per explant [44].

### 2.3. Essential Oils

In vitro microbial contamination is one of the most serious problems when culturing tissue. Using plant extracts such as essential oils for explant disinfection and establishing an aseptic culture medium to replace autoclaving was found to be an alternative procedure for plant tissue culture [58]. Oxidative browning from the accumulation and oxidation of polyphenols in the media is another impediment to in vitro propagation. For the successful establishment of in vitro tissue cultures, it is necessary to limit the oxidation of phenolics, the source of enzymatic browning, and inhibit microbial growth [45]. In the study of Hamdeni et al. [45], *Rosmarinus officinalis* and *Thymus vulgaris* essential oils were assayed for their effectiveness in controlling enzymatic browning and contamination of cultures from *Aloe vera*. While *T. vulgaris* essential oil inhibited explant growth, *R. officinalis* induced the highest explant survival percentage (100%) with no signs of browning after four weeks of culture with concentrations of 0.05%, 0.075%, and 0.1%. The lowest infection percentage (10%) was observed for media containing 0.075% and 0.1% of *R. officinalis*. The highest average number of leaves per explant was 3.71 with 0.1% *R. officinalis* and the greatest leaf length was 3.18 cm with 0.05%. According to Taghizadeh et al. [46], sterile medium conditions were obtained by using eugenol, carvacrol, or thymol at 0.01% and 0.5% which inhibited the growth of fungi and bacteria contaminations respectively with no autoclaving of the medium and vessels. The inhibitory effect of essential oils from *Mentha piperita*, *Thymus vulgaris*, and *Cinnamomum camphora* against common fungal contamination affecting the tissue culture of *Phoenix dactylifera* was reported by Jasim et al. [47]. These essential oils at 2% each resulted in 100% inhibition of mycelium growth of fungi species (*Alternaria* spp., *Fusarium* spp., and *Aspergillus* spp.) compared to the control treatment (10% of fungal contamination) with no disinfecting agent. The disinfecting properties of thymol and carvacrol (at 200 mg/L each and for 60 to 120 min exposure time) led to the appropriate control of fungi and bacterial infection of *Cynodon dactylon* nodal explants [48].

## 3. Marker-Assisted Genetic Stability Assay

The unsustainable use of natural resources has harmful consequences that threaten biodiversity and multiple ecosystem services and negatively affect human well-being. Promoting conservation is recommended to halt the erosion of genetic resources and habitats and maintain ecosystem functions [59]. Where preserving the original gene pool is of primary importance, micropropagated plant materials are an appropriate tool for biodiversity restoration (conservation, re-introduction, and recovery) of rare, endangered, and threatened plant species [60,61]. Even though plant regeneration from organized meristems (shoot tips and axillary buds) is a reliable way to produce true-to-type plants compared with callus and somatic embryogenesis regeneration, in vitro culture environments can be mutagenic [62,63,64]. DNA methylation, point mutations, and chromosome rearrangements are the major causes of somaclonal variation, which arises due to in vitro stresses such as artificial lighting, nutrient compounds [64,65], plant growth regulators [66,67,68], explant source and genotype, culture duration, and subculture number [62,63,69,70,71,72,73,74]. Hussien et al. [64] reported that the interaction between sucrose and minerals levels led to somaclonal variation (38.33%) among *Populus alba* propagated plants. The supplementation of high concentrations of copper sulphate to the MS medium showed polymorphism among *Musa* sp. in vitro regenerated plants when treated with 60 mg/L with an average of 1.8% [65]. Bhattacharyya et al. [66] indicated that plant growth regulators (BA and meta-topolin) showed clonal variability at various stages of sub-culturing as well as after successful acclimatization due to their residual toxicity which led to DNA damage and microsatellite instability within the micropropagated plantlets of *Ansellia africana*. A low level of genetic polymorphism of *Lilium davidii* occurred when BA and TDZ cytokinins were supplemented to the culture media [67]. The highest genetic variability (16%) of *Rhododendron* ‘Kazimierz Odnowiciel’ was found for both BA and TDZ compared to the other cytokinins combinations [68]. The maintaining of cultures for a long period and callus regeneration from *Spondias pinnata* resulted in slight polymorphism (12.5%), which may be due to the stress of in vitro culture conditions [63]. Micropropagated plantlets and callus-derived regenerants from *Camellia sinensis* showed molecular changes after 7 years of in vitro propagation with polymorphism percentages ranging from 43% to 67% [73]. Genetic variation occurred among two genotypes (Mississippi and SWD) of *Anthurium andraeanum* after callus induction and long-term maintenance [74]. Therefore, it is necessary to confirm true-to-type propagules at an early stage of development. Molecular, biochemical, and morphological analyses are the main approaches for determining the clonal stability of in vitro-generated plantlets [75,76,77,78,79]. PCR-based molecular markers such as random amplified polymorphic DNA (RAPD) have been found to be enormously helpful in ascertaining the genetic stability of in vivo cultivated as well as in vitro regenerated plants [80,81]. Although RAPD primers have been successfully used for assessing the genetic stability of several species, they have a number of limitations such as reproducibility and dominance [82,83]. Consequently, using a different molecular marker system is required to provide a more reliable result.

Recently, many DNA molecular markers have been developed to confirm genetic stability. Among them, the most advanced and reliable with high reproducibility rates are inter-simple sequence repeats (ISSR) and start codon targeted (SCoT) [84]. Simple sequence repeat (SSR) markers have also gained considerable importance in genetic stability assessment due to their high reproducibility, co-dominant inheritance nature, sensitivity, and strong discriminatory power [85]. A promotional effect of organic growth additives (amino acids, polyamines, proteins, plant, and fruits extracts) in stimulating cells proliferation, morphogenesis, and tissues development was reported by several authors, but the genetic stability of in vivo cultivated as well as in vitro regenerated plants needs to be ascertained [86,87,88] (Table 2).

### 3.1. RAPD Analysis

The optimum development and maturation of somatic embryos from *Sterculia alata* were observed by the supplementation of 400 mg/L Gln to MS medium as a nitrogen source. The genetic analysis using 25 RAPD primers produced a total number of 181 amplification bands with an average of 7.24 bands per primer. The amplification profiles generated the same banding pattern in all the samples confirming that the supplementation of amino acids to the in vitro propagation protocol is reliable in producing true-to-type plants [89]. Thakur et al. [93] found that adding 10 mg/L of CH to the MS medium resulted in the best multiplication rate (5.08) and shoot elongation with an average shoot length of 2.94 cm of *Prunus salicina*. Out of 16 RAPD primers tested, 14 primers produced 43 amplification fragments ranging from 200 to 1500 bp in size. All banding profiles were monomorphic across all the tested plants and similar to those of the mother plant. The work of Al-Mayahi [90] indicates that adding polyamines (PU and Spd) to MS medium of *Phoenix dactylifera* is beneficial in producing genetically stable plants. All four RAPD primers showed amplifications with monomorphic bands among both in vitro-derived plants and the mother plant. The two types of polyamines at 75 mg/L each were the most effective treatments in root formation and number. Gurme et al. [91] reported that when 15% CW was supplemented to MS medium, the best shooting frequency (90%) and plant numbers (18) from *Amorphophallus paeoniifolius* were obtained. Among 10 RAPD primers tested, only 6 generated a total of 292 bands without any polymorphic bands between 10 in vitro regenerants and their mother plant. Ten RAPD primers were used to evaluate the genetic profile of *Rhynchostylis retusa* tissue cultured plantlets developed under CW and fungal elicitor treatments. The highest shoot numbers and lengths were found on MS medium supplemented with 10% CW while the fungal elicitor showed the best response for root number and length. All RAPD primers produced 23 amplification bands ranging in size from 275 to 1100 bp. The genetic stability among in vitro cultured plants and the mother plant was maintained [88]. Adding 1% AvLE to MS liquid media fully controlled culture contamination and enhanced shoot numbers and lengths of *Bambusa balcooa*. Out of 20 RAPD primers, only 8 yielded 22.44 reproducible and scorable bands with 2.8 bands per primer ranging in size from 100 to 1800 bp. All amplified bands were monomorphic revealing that *Bambusa balcooa* plants retained their genetic stability [92].

### 3.2. ISSR Analysis

Nandhakumar et al. [94] reported that 10 ISSR primers were used to assess the genetic stability of *Musa* spp. (Cultivars Grand Naine and Rasthali) when Pro and Gln were added to MS media. Gln (400 mg/L) significantly enhanced the number of both primary (1680, 1850) and secondary (3597, 3270) somatic embryos per culture from Grand Naine and Rasthali cultivars, respectively. ISSR primers generated a total of 1534 and 1488 bands ranging from 200 bp to 2500 bp in size in Grand Naine and Rasthali cultivars, respectively, giving rise to only monomorphic bands across all the tested plants in both cultivars.

### 3.3. SSR Analysis 

Natarajan et al. [87] found that the highest somatic embryo induction (85%) was observed when CH (100 mg/L) and Gln (150 mg/L) were supplemented to the nutrient media of *Musa* AAB cultivar Chenichampa as compared to the control. No somaclonal variations were detected using 10 SSR primers among the embryogenic cell suspension-derived plants and their mother plant. Asadi-Aghbolaghi et al. [95] reported that the highest embryogenic callus induction was observed when CH (100 mg/L) and Gln (500 mg/L) were supplemented to the culture media of *Stipagrostis pennata*. Ten SSR primes were tested, out of which four showed a single amplification band size of 185, 412, 243, and 210 bp for primers 1, 3, 7, and 8, respectively. The regeneration protocols adding both CH and Gln as organic growth additives showed genetic stability using SSR primers which makes these protocols reliable.

### 3.4. Combined Markers Analysis (RAPD, ISSR, and SCoT)

For better genetic stability analysis, it is highly recommended to use a combination of more than one type of molecular marker that amplifies different regions of the plant genome.

#### 3.4.1. RAPD and ISSR Analysis

RAPD and ISSR analysis confirmed the genetic stability of both *Valeriana jatamansi* [96] and *Ranunculus wallichianus* [97] in vitro regenerated plants on MS medium supplemented with 10% CW. For *Valeriana jatamansi* propagation protocol, CW supplementation resulted in the maximum response with regard to shoot and root numbers and lengths (13, 19.6, 6, and 7.5 cm), respectively. Out of 35 RAPD and 10 ISSR primers analyzed, only 10 RAPD and 5 ISSR primers produced a total of 32 and 12 similar banding patterns between in vitro raised plantlets and the mother plant, respectively [96]. For *Ranunculus wallichianus*, adding CW to MS medium resulted in the highest regeneration response (97%), the number of shoots formed (11.1 shoots/culture), and shoot length (9.2 cm). Nine RAPD and eight ISSR primers produced 56 and 47 bands with an average of six and five bands per primer ranging from 200 to 1500 bp and 200 to 1000 bp in size, respectively [97]. Krishnan et al. [86] approved the significant efficacy of incorporating plant extracts such as SSE into the culture medium. The SSE supplementation in a dose-dependent manner resulted in the highest shoot and root length and root biomass. Only two RAPD (out of six) and four ISSR (out of eleven) primers produced stable amplicons with 11 and 26 monomorphic amplicons, respectively.

#### 3.4.2. RAPD and SCoT Analysis

RAPD and SCoT markers proved the genetic stability of both *Citrullus lanatus* [98] and *Pisum sativum* [99] in vitro-raised plants when the two polyamines (Spd and PU) were supplemented to their nutrient media. Spd and PU at 10 mg/L each, increased the shoot induction response (93%), shoot number (46.43 shoots/explant), and shoot elongation (6.3 cm) for *Citrullus lanatus*. The highest rooting percentage (95%) with the production of 23.03 roots per shoot measuring 4.32 cm in length was also recorded. In total, 9 RAPD and 17 SCoT primers produced 41 and 47 monomorphic fragments in the size range of 200 to 1800 and 300 to 2000 bp, respectively [98]. However, Spd at 20 mg/L resulted in the highest multiple shoot numbers (65.1 shoots/explant) while PU at 30 mg/L produced the highest number of roots (33.66 roots/shoot) for *Pisum sativum*. Nine RAPD and 17 SCoT primers produced 34 and 38 monomorphic fragments with the ranges of 400 to 600 and 100 to 500 bp, respectively [99].

#### 3.4.3. ISSR and SCoT Analysis

Both ISSR and SCoT markers certified the genetic stability of *Helicteres isora* regenerated on MS medium supplemented with Gln. At 50 mg/L concentration, it produced the highest shoot numbers (21.3, 16.9) from both cotyledonary node and axillary node explants, respectively. Five ISSR primers (out of ten) produced 27 reproducible bands with 5.4 bands per primer varying in size from 0.3 to 2.2 kb. Out of the 17 SCoT primers tried, 13 produced 63 monomorphic bands with 4.8 bands per primer ranging in size from 0.3 to 3 kb. In both ISSR and SCoT techniques, all the resolved bands were monomorphic to all in vitro regenerated plants as well as their in vivo-based mother plant [100].

## 4. Conclusions

The most recent results summarized in this review show that micropropagation techniques can be enhanced by supplementing organic growth additives for producing genetically stable plants to meet the increasing demand for planting raw materials. However, it is extremely important to understand the structure-mode of action relationship of these supplements. In this regard, mutual cooperation between biotechnologists, chemists, and biologists may play a major role in proposing adequate growth additives for each plant species. That strategy could be an efficient tool for the rehabilitation of degraded landscapes and biodiversity restoration via the large-scale production of selected genotypes of medicinal, pastoral, and forest species.

## Figures and Tables

**Table 1 plants-11-03087-t001:** Effect of different organic growth additives on in vitro tissue culture.

Type of Nutrients	Species	Explant	Culture Media	Organic Growth Additives	Main Results	References
**Vegetable, fruit, and plant extracts**	*Daphne* sp. (*caucasica*, *tangutica*, *jasminea*)	Shoot	MS	CW: 10 mL/L PP: 10 mL/L AAN: 20 mg/L CHT: 15 mg/L *Desmodesmus subspicatus*: 20%, 50%	*D. tangutica*: Shoot proliferation was improved by medium supplementation with CW and PP (micropropagation coefficient 16.6 and 13.4). *D. caucasica*: The highest frequency of adventitious rhizogenesis (57.1%) occurred when tissues were cultured on MS + 10 mL/L PP. *D. jasminea:* The rooting percentage was 14% and 8.9% for PP and CW.	[24]
	*Celosia* sp.	Stem	MS	MCJ: 20, 30, 50, 70 mL/L YCJ: 20, 30, 50, 70 mL/L PJ: 20, 30, 50, 70 mL/L BJ: 20, 30, 50, 70 mL/L TJ: 20, 30, 50, 70 mL/L	YCJ at 70 mL/L induced the highest shoot regeneration (14.21). BJ and TJ promote the highest shoot regeneration of stem segments at 50 mL/L which produced 9.57 and 9.28 shoots per explant, while PJ, at the lowest concentration (20 mL/L), showed the highest shoot regeneration (10.5).	[25]
	*Raphunus sativus*	Hypocotyl	MS	CW: 20%, CM: 20%, GSL: 10%, GPT: 10%, GC: 10%, RF: 5%, GG: 10%, GP: 10%, BF: 10%, OJ: 10%	The highest number of shoots (12 shoots/explant) was observed in MS supplemented with 2.5 mg/L BAP + 0.1 mg/L NAA + 10% OJ, whereas 8 shoots/explant were obtained with 20% CW. Media with GSL, RF, GG, GPT, and BJ inhibit shoot regeneration.	[14]
	*Epidendrum nocturnum*	PBLs	OSM	BW: 10 g/L CW: 10 mL/L 5 g/L BW + 5 mL/L CW PD: 10 g/L	Media with 5 g/L BW + 5 mL/L CW showed greater plant length (19.80 mm), the number of roots (2.1), and fresh weight (0.08 g).	[16]
	*Cypripedium macranthos*	PLBs	¼ MS	CW: 50, 100, 200 mL/L BSP: 50, 100, 200 mL/L MSP: 50, 100, 200 mL/L BW: 15, 30, 60 g/L P: 1, 2, 4 g/L	The highest germination and protocorm formation percentages (70.8% and 74.2%) were obtained with 100 mL/L CW.	[15]
	*Dendrobium lowi*	Protocorms	KC	CW: 10%, 15%, 20% TJ: 10%, 15%, 20% BP: 25, 75, 125 g/L P: 2 g/L	Protocorm treated with 25 g/L BP showed the highest GI values of 593.3 with 100% protocorms successfully developing shoots and 93.3% of protocorms producing root.	[26]
	*Musa* cv. Rajabulu	Shoots	MS	CW: 50, 100, 150, 200 mL/L	The supplementation of CW (50 and 100 mL/L) showed the best results on the average number of roots (9.33, 9.00) and root length (11.6, 10.76 cm). Acclimatization succeeded (100%) with CW (50 to 200 mL/L).	[27]
	*Musa acuminata*	Shoot	MS	CW: 200 mL/L PPJ: 200 mL/L OJ: 200 mL/L	MS supplemented with 200 mL/L CW resulted in the highest number of shoots/explant (1.71), the longest shoot length (4.25 cm), and the highest number of leaves/explant (4).	[18]
	*Dendrobium chryseum*	Protocorms	½ MS	CW: 5%, 10%	The highest number of shoots developed on ½ MS fortified with 2 mg/L Kn + 10% CW and the longest shoots were obtained on ½ MS + 1 mg/L GA_3_ + 10% CW.	[28]
	*Dianthus caryophyllus*	Shoot tip Node	MS	CW: 5%, 8%, 10%, 15%, 20%	The best regeneration was obtained on MS supplemented with 1 mg/L BAP +10% CW which increased the number of shoots per culture (nodal explant:113.83 and, shoot tip explant: 93.33).	[29]
	*Gastrochilus matsuran*	PLBs	½ MS	CW: 0%, 2.5%, 5%, 7.5%, 10%	When 5% CW was added to ½ MS + 0.05% AC + 1% BP + 0.2% P + 1 µM NAA + 1.5 µM GA3. It produced the highest germination percentage at 93.3%.	[30]
	*Hylocereus polyrhizus*	Stem	MS	MCW: 2%, 4%, 6%	Adding 4% MCW and 3% sucrose in MS media increased the shoot elongation (2.45 cm).	[31]
	*Echinacea purpurea*	Petiole	MS	LH: 100, 300, 900 mg/L P: 100, 300, 900 mg/L Y: 100, 300, 900 mg/L CW: 2%, 4%, 8%	Shoot multiplication has been enhanced with 2% CW on MS containing 0.3 mg/L BA + 0.01 mg/L NAA (2.58 buds/explant).	[32]
	*Physalis angulata*	Node	MS	TJ: 5%, 7.5%, 10% B: 1.25%, 2.5%, 3.75%	The maximum shoot number (12.5) was produced in MS + 2 mg/L BAP + 0.05 mg/L IAA + 5% TJ while the maximum shoot length (10.7 cm) was obtained with 7.5% TJ.	[33]
	*Laelia rubescens*	PLBs	Phy MS ½ MS	CW: 100 mL/L PJ: 100 mL/L	Phy media supplemented with 100 mL/L PJ resulted in the highest seedling height (1.31 cm), number of leaves per seedling (3.33), and roots (2.33)	[34]
	*xLaeliocattleya*	PLBs	OSM MS	CW: 200 mL/L PJ: 200 mL/L	MS supplemented with PJ produced the highest results in asymbiotic germination (56%) and seedlings formation (25.8).	[35]
	*Aloe vera*	Shoot tip	Shoot multiplication: MS	AvG: 2.5%, 5%, 10%	Adding AvG to MS media increased the number of axillary shoots compared to the control. The highest axillary shoot number was recorded on a medium containing 5% AvG (13.27) and the highest shoot elongation (2.5 cm) was recorded on a medium supplemented with 2.5% AvG.	[36]
			Rooting: ½ MS	AvG: 10%, 20%, 30%	The highest root number and the greatest root length (5.73 and 5.90 cm) were recorded on MS medium supplemented with 10% AvG.	
	*Aloe vera*	Rhizomatous stem	Shoot multiplication: MS	AvG: 5%, 10%, 15%, 20%	The highest number of formed shoots per explant (length ≥ 2 cm) was 17.8 shoots on MS medium supplemented with 10% AvG.	[37]
			Rooting: ⅓ MS	AvG: 10%, 20%, 30%, 40%	Adding AvG (20% and 30%) to ⅓ MS increased the rooting response to 100%, the number of roots per shoot (9.8 and 9.2), and the length of the roots (3.1 and 2.8).	
	*Bacopa chamaedryoides*	Shoot tip, nodes	Rooting: ½ MS	AvG: 50%	The best rooting response (100%), number (18.3), and length of shoots (2.3 cm) were achieved on ½ MS supplemented with 50% AvG.	[38]
**Amino acids, polyamines, and proteins**	*Aegle marmelos*	Nodes, internodes, shoot tip, leaves	MS	Amino acids (Met, Glu, Gly, Trp, Pro, Lys, Arg, Gln): 10, 20, 30, 40 mg/L	MS medium containing 2 mg/L BAP and a combination of all amino acids in 20 mg/L obtained the optimum axillary shoot proliferation response (100%) and average shoot numbers per explant (2.22).	[39]
	*Chlorophytum borivilianum*	Node	Shoot morphogenesis: MS	Ads: 10, 20 mg/L CH: 10, 20, 30 mg/L PU: 0, 10, 30, 50, 70, 100 mg/L	The best response for shoot morphogenesis (96.67%) was achieved on MS supplemented with 2 mg/L BAP + 1 mg/L NAA + 20 mg/L Ads.	[40]
			Rooting: ½ MS	PU: 0, 3, 6, 9, 12, 18 mg/L	½ MS with IBA (2 mg/L) + PU (9 mg/L) was best for rooting with 83.33% root induction.	
	*Rosa canina*	Node	VS	CH: 200, 400, 600 mg/L Glu: 2, 4, 8, 12 mg/L Pro: 500, 1000, 1500, 2000 mg/L	The maximum shoot number (4.1 shoots/explant) was obtained in VS supplemented with 600 mg/L CH. Glu at 12 mg/L enhanced shoot regeneration and leaf number compared to the control.	[41]
	*Orthosiphon aristatus*	Node	MS	P: 50, 100, 150, 200 mg/L, Asn: 10, 20, 30, 40 mg/L Gln:10, 20, 30, 40 mg/L WJ: 5%, 10%, 15%, 20% CW: 5%, 10%, 15%, 20% CW (10%) + Gln: 20, 30 mg/L CW (10%) + Asn: 20, 30 mg/L	MS supplemented with 1 mg/L BAP + 0.5 mg/L KIN + 10% CW and Gln or Asn increased the number of shoots/explant. Gln at 30 mg/L produced 44.04 shoots/explant with a mean length of 7.47 cm, whereas 20 mg/L Asn resulted in 40.43 shoots/explant and 6.89 cm shoot length after 8 weeks of culture.	[42]
	*Vanda helvola*	Protocorms	KC	TJ: 10%, 20%, 40% CW: 10%, 20%, 40% P: 0.1%, 0.2%, 0.4% YE: 0.1%, 0.2%, 0.4%	Protocorms (99.50%) treated in KC containing 0.1% P successfully produced 3.10 leaves with an average length of 10.97 mm per responsive explant after 90 days of culture.	[43]
	*Stevia rebaudiana*	Node	MS	CH: 0.025%, 0.05%, 0.075%, 0.1% CW: 5%, 10%, 15%, 20% ME: 0.025%, 0.05%, 0.075%, 0.1% YE: 0.025%, 0.05%, 0.075%, 0.1%	MS supplemented with 2 mg/L BAP + 0.5 mg/L Kin + 0.1 mg/L NAA + 0.05% CH resulted in a maximum of 15 shoots. A 90% regeneration frequency and shoot length of 6 cm were recorded per explant.	[44]
**Essential oils**	*Aloe vera*	Shoot tip	MS	AC: 0.5 g/L *T. vulgaris* EO: 0.1%, 0.2% *R. officinalis* EO: 0.025%, 0.05%, 0.075%, 0.1%	Explant survival was 100% after 4 weeks of culture with *R. officinalis* EO concentrations of 0.05%, 0.075%, and 0.1% with no signs of browning. The lowest infection percentage (10%) was observed for media containing 0.075% and 0.1% of *R. officinalis* EO. The highest number of leaves per explant was 3.71 with 0.1% *R. officinalis* EO and the greatest leaf length was 3.18 cm with 0.05%.	[45]
	*Fragaria* × *ananassa* (Duch)	Leaves	MS	Eugenol: 0.01%, 0.02%, 0.04%, 0.5%, 2.5%, 5% Carvacrol: 0.01%, 0.02%, 0.04%, 0.5%, 2.5%, 5% Thymol: 0.01%, 0.02%, 0.04%, 0.5%, 2.5%, 5%	All essential oil treatments resulted in sterile conditions of the medium. The growth of in vitro contaminations from fungi and bacteria was inhibited at 0.01% and 0.5% concentrations, respectively.	[46]
	*Phoenix Dactylifera*	Node	MS	*Mentha piperita*: 2% *Thymus vulgaris*: 2% *Cinnamomum camphora*: 2%	All the essential oils inhibited the mycelial growth and fungus contamination of tissue culture.	[47]
	*Cynodon dactylon*	Node	MS	Thymol: 100, 200 mg/L (30, 60, and 120 min soaking time) Carvacrol:100, 200 mg/L (30, 60, and 120 min soaking time)	Increasing the period of exposure (60–120 min) with thymol and carvacrol at 200 mg/L led to the appropriate control of fungi and bacterial infection of explants.	[48]

**Table 2 plants-11-03087-t002:** Molecular markers used for genetic stability assessment of in vitro propagated plants under organic growth addition.

Molecular Marker	Species	Organic Growth Additives	Main Results	References
RAPD	*Sterculia alata*	Pro: 50, 100, 200, 300, 400 mg/L Gln: 100, 200, 300, 400, 500, 600 mg/L	The optimum development and maturation of somatic embryos were observed by the supplementation of 400 mg/L Gln. All 25 RAPD primers generated distinct amplification profiles with the same banding pattern of all the samples. The total number of amplification products was 181 bands with an average of 7.24 bands per primer.	[89]
	*Phoenix dactylifera*	PU: 25, 75, 150 mg/L Spd: 25, 75, 150 mg/L	The two types of polyamines (PU and Spd) at 75 mg/L each, were the most effective treatments in root formation and number. All four RAPD primers showed unambiguous amplifications with monomorphic bands among both in vitro-derived plants and the mother plant.	[90]
	*Amorphophallus paeoniifolius*	CW: 15% Gln: 1.36, 3.42 µM CH: 0.05%	MS medium supplemented with 15% CW in combination with 4.43 µM BAP resulted in the best shooting frequency (90%) and plant number (18). Among 10 RAPD primers tested, only 6 generated a total of 292 bands without any polymorphic bands between 10 in vitro regenerants and their mother plant.	[91]
	*Bambusa balcooa*	AvLE: 1%	The addition of 1% AvLE to MS liquid media has fully controlled culture contamination and enhanced shoot number and length. Out of 20 RAPD primers, only 8 primers yielded 22.44 reproducible and scorable bands with 2.8 bands per primer ranging in size from 100 to 1800 bp. All amplified bands were monomorphic across the in vitro-raised plants and their mother plant.	[92]
	*Rhynchostylis retusa*	CW: 5%, 10% Fungal elicitors isolated from *Vanda cristata*	The highest shoot number and length were found on MS medium supplemented with 10% CW, while a fungal elicitor showed the best response for root number and length. All RAPD primers (10) produced 23 amplification bands ranging in size from 275 to 1100 bp. Genetic uniformity among in vitro cultured plants and the mother plant was maintained.	[88]
	*Prunus salicina*	CH: 10, 50, 100 mg/L	CH (at 10 mg/L) resulted in the best multiplication rate (5.08) and shoot elongation with an average shoot length of 2.94 cm. Out of 16 RAPD primers tested, 14 primers produced 43 amplification fragments ranging from 200 to 1500 bp in size. All banding profiles were monomorphic across all of the tested plants and similar to those of the mother plant.	[93]
ISSR	*Musa* spp. (Cultivars Grand Naine and Rasthali)	Pro: 100, 200, 300, 400 mg/L Gln: 100, 200, 300, 400 mg/L Asn: 50, 100, 150, 200 mg/L	Gln (400 mg/L) significantly enhanced the number of both primary (1680, 1850) and secondary (3597, 3270) somatic embryos per culture from Grand Naine and Rasthali cultivars, respectively. All 10 ISSR primers generated a total of 1534 and 1488 bands ranging from 200 bp to 2500 bp in size in Grand Naine and Rasthali cultivars, respectively, giving rise to only monomorphic bands across all the tested plants in both cultivars.	[94]
SSR	*Musa* AAB cultivar Chenichampa	ME: 100 mg/L YE: 100 mg/L CH: 50, 100, 150 mg/L Gln: 50, 100, 150 mg/L	The highest somatic embryo induction (85%) was observed when CH (100 mg/L) and Gln (150 mg/L) were supplemented to the media as compared to the control. No somaclonal variations were detected using 10 SSR primers among the embryogenic cell suspension-derived plants and their mother plant.	[87]
	*Stipagrostis pennata*	Gln: 500 mg/L CH: 100 mg/L	The highest embryogenic callus induction was observed when Gln and CH were supplemented to the culture media. In total, 10 SSR primes were tested, out of which 4 primers showed a single amplification band size of 185, 412, 243, and 210 bp for primers 1, 3, 7, and 8, respectively.	[95]
RAPD and ISSR	*Oryza sativa*	SSE: 10%, 20%, 30%, 40%	The SSE supplementation in a dose-dependent manner resulted in the highest shoot, root length, and root biomass. Only 2 RAPD (out of 6) and 4 ISSR (out of 11) primers produced stable amplicons with 11 and 26 monomorphic amplicons, respectively.	[86]
	*Valeriana jatamansi*	CW: 10%	The supplementation of 10% CW resulted in the maximum response with regard to shoot and root numbers and lengths (13 cm, 19.6 cm, 6 cm, and 7.5 cm, respectively). Out of the 35 RAPD and 10 ISSR primers analyzed, only 10 RAPD and 5 ISSR primers produced a total of 32 and 12 similar banding patterns between the in vitro raised plantlets and the mother plant, respectively.	[96]
	*Ranunculus wallichianus*	CW: 10%	MS medium supplemented with 10% CW resulted in the highest regeneration response (97%), number of shoot formations (11.1 shoots/culture), and shoot length (9.2 cm). In total, 9 RAPD and 8 ISSR primers produced 56 and 47 bands with an average of 6 and 5 bands per primer ranging from 200 to 1500 bp and 200 to 1000 bp in size, respectively.	[97]
RAPD and SCoT	*Citrullus lanatus*	Spd: 5, 10, 15, 20, 25 mg/L Spm: 5, 10, 15, 20, 25 mg/L PU: 5, 10, 15, 20, 25 mg/L	Spd (10 mg/L) increased shoot induction response (93%), shoot number (46.43 shoots per explant), and shoot elongation (6.3 cm). PU (10 mg/L) showed the highest rooting percentage (95%) with the production of 23.03 roots per shoot measuring 4.32 cm in length In total, 9 RAPD and 17 SCoT primers produced 41 and 47 monomorphic fragments in the size range of 200–1800 and 300 to 2000 bp, respectively.	[98]
	*Pisum sativum*	Spd: 5, 10, 15, 20, 25, 30, 35 mg/L Spm: 5, 10, 15, 20, 25, 30, 35 mg/L PU: 5, 10, 15, 20, 25, 30, 35 mg/L	The highest multiple shoots number (65.1 shoots/explant) was attained with 20 mg/L SPD while 30 mg/L PU produced the highest number of roots (33.66 roots/shoot). In total, 9 RAPD and 17 SCoT primers produced 34 and 38 monomorphic fragments with the ranges of 400 to 600 and 100 to 500 bp, respectively.	[99]
ISSR and SCoT	*Helicteres isora*	Gln: 25, 50, 75, 100 mg/L SC: 10, 20, 30, 40 mg/L	Gln (50 mg/L) produced the highest shoot number (21.3, 16.9) from both cotyledonary node and axillary node explants, respectively. In total, 5 ISSR primers (out of 10), produced 27 reproducible bands with 5.4 bands per primer varying in size from 0.3 to 2.2 kb. Out of the 17 SCoT primers tried, 13 primers produced 63 monomorphic bands with 4.8 bands per primer ranging in size from 0.3 to 3 kb. In both ISSR and SCoT techniques, all the resolved bands were monomorphic to all in vitro regenerated plants as well as their in vivo-based mother plant.	[100]

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
