# Peer review of "Incorporation of Organic Growth Additives to Enhance In Vitro Tissue Culture for Producing Genetically Stable Plants"

_plants, 2022, doi:10.3390/plants11223087_

Round 1
Reviewer 1 Report
The manuscript entitled ‘Incorporation of organic additives to enhance in vitro tissue culture for producing genetically stable plants’ is a good manuscript with ample information on the use of organic additives for cost effective tissue culture media. The author has done a commendable job for gathering, and processing the data as per the requirement of the manuscript title. However, the manuscript requires polishing. Below are the comments that can be considered to improve further:
Comments
· Grammar, verb agreement and punctuations errors are often seen
· Avoid redundancy of the sentences
· Units: maintain uniformity in the writing units, e.g. mg L-1 or mg/l
· Either use common name or botanical names, follow the same across the manuscripts
· Kindly check the whole manuscript for such errors
1. Line 71: correct font
2. Line 98: [21] reported the successful…….., can be written as ‘Author et al., 2013……… ….. ……………[21].’ For the ease of audience/readers to figure out recent studies. Do not forget to cite references at the end of the sentence
3. Line 153: antioxydant spelling
4. Line 152-154: provide citation
5. Line 160-162; 169-170; 178-179: Author says other studies, however cited only one reference.
6. Line 172-173: author has mentioned common name and botanical names together, please use anyone and follow the same across the manuscripts
7. Line 217-218: sentence starting from Climate change already used in introduction
8. Line 217-220: Introductory sentences, can be removed from the core topic/section
9. Line 242: New paragraph can be started
Author Response
Dear Reviewer,
Thank you for your precious review.
Please you find below our point-by-point responses to your recommendations:
The manuscript entitled ‘Incorporation of organic additives to enhance in vitro tissue culture for producing genetically stable plants’ is a good manuscript with ample information on the use of organic additives for cost effective tissue culture media. The author has done a commendable job for gathering, and processing the data as per the requirement of the manuscript title. However, the manuscript requires polishing. Below are the comments that can be considered to improve further:
Comments
- Grammar, verb agreement and punctuations errors are often seen
Grammar, verb agreement and punctuations errors were corrected across the manuscript.
- Avoid redundancy of the sentences
Redundant sentences were deleted.
- Units: maintain uniformity in the writing units, e.g. mg L-1 or mg/l
Uniformity in writing the units was maintained across the manuscript.
- Either use common name or botanical names, follow the same across the manuscripts
Botanical names were used across the manuscript.
- Kindly check the whole manuscript for such errors
- Line 71: correct font
Font was corrected.
- Line 98: [21] reported the successful…….., can be written as ‘Author et al., 2013……… ….. ……………[21].’ For the ease of audience/readers to figure out recent studies. Do not forget to cite references at the end of the sentence
Done across the manuscript.
- Line 153: antioxydant spelling
Antioxydant was replaced by antioxidant.
- Line 152-154: provide citation
Citations were added.
- Line 160-162; 169-170; 178-179: Author says other studies, however cited only one reference.
Done.
- Line 172-173: author has mentioned common name and botanical names together, please use anyone and follow the same across the manuscripts
Botanical names were used across the manuscript.
- Line 217-218: sentence starting from Climate change already used in introduction
The sentence was deleted.
- Line 217-220: Introductory sentences, can be removed from the core topic/section
Introductory sentences were deleted.
- Line 242: New paragraph can be started
Done.
Thank you again for your consideration and service.
Sincerely,

Reviewer 2 Report
The review manuscript entitled “Incorporation of organic additives to enhance in vitro tissue culture for producing genetically stable plants” contains interesting information on summarized data of the application of non-conventional organic supplements in plant cell tissue and organ culture. As such it would be of great interest to the audience. My recommendation is to encourage authors to resubmit after drastic revision of the work following the remarks below.
The main conclusions of the presented review, that non-conventional additives are superior to conventional ones in terms of preservation of genetic stability of in vitro cultured plants lie on the subjective correlation of two groups of non-related datasets.
Thus, Table 2 presents data on a bibliographical survey of a number of examples of genetic variation cases in conventional tissue culture and Table 3 summarizes cases of research of other authors in which non-conventional organic supplements were added to the culture media and no or only minor genetic alterations were established.
In order to make a sound conclusion, comparison has to be made in one and the same experiment. In the case of the present manuscript, please try to change the focus of summary of data in order to keep the information of both Table 2 and Table 3. Or, which I would recommend, keep only a few examples of Table 2 as a text, but strongly expand and improve the manuscript in a different aspect of serious scientific representation of what non-conventional organic supplements are and keep only the data of Table 3.
Conclusion
The concretely reviewed bibliography in the manuscript does not provide examples of solving land degradation and desertification problems. The accent of the work is different and the accent of the conclusion has to be related to the main idea of the work. The issue with land degradation has also been used in supporting the review topic above. Please, change the focus of the work according to its content.
In addition, below are only a few non-exclusive concrete remarks on the scientific writing of the manuscript.
Title
The term “organic additives” standing in the title does not represent concretely what types of organic additives are being reviewed in the work.
Abstract
In the Abstract and everywhere throughout the text: the expression “true-to-true type” has to be corrected with “true-to-type”.
The whole abstract has to be modified in accordance with the modified content of the re-worked manucript.
“Organic substances”: the same remark as the title. The term has to be substituted with a more concrete and informative one. Conventionally used plant growth regulators, sucrose, vitamins are also organic substances.
Also application of amino acids (for example glycine) is not a non-conventional approach such as fruit juice or coconut water, etc.
Introduction
Expressions such as “several ecosystems” are not informative and not concrete.
Line 67and 70 and elsewhere throughout the text: “synthetic”, “synthetic materials”: The conventionally used tissue culture media ingredients are not necessarily always synthetic.
Chapter 2 “Supplementation of organic additives…..”: the main point of comparison of conventional vs. non-conventional organic additives in tissue culture is not that the first ones are synthetic and the second group - natural. They differ principally in their origin and chemical content. This has to be very well and scientifically-based presented in the manuscript. The reason that the non-conventional organic additives DO WORK in tissue culture is that they contain certain components which serve the SAME functions as of the conventional tissue culture media components (auxins, cytokinins, vitamins, etc). Please expand the review by adding scientifically based discussion on the chemical content and mechanism of action of the non-conventional organic tissue culture additives which are being discussed in the present work.
Everywhere throughout the text: (1) Please, arrange references chronologically when in brackets, (2) References have to be numbered according to their appearance in the text. Please follow the guidelines of the Journal.
Table 1: Please, insert a system to define the order of records in Table 1: are they arranged according to the alphabetical order of species/component/other, or according to the type of organic supplement applied, or the type of tissue culture or the type of effect achieved or other? Does the table follow the numbers of grouping the supplements in 2.1, 2.2, etc?
Below tables it is written “Author’s construction”. Table presentation of data should always be author's construction; otherwise reprinting a table from another source would be a case of plagiarism. So, it is not necessary to explicitly mention “Author’s construction”.
The respective Tables in which the discussed information is contained have to be cited throughout the text.
2.3. Essential oils: Please, specify for the different cases if essential oils were included as components in the medium and the explants - cultivated in their presence? Or were they used as initial sterilizing agents instead of conventional sterilizing agents in tissue culture initiation.
For example in reference [42], the essential oils were included in media after cultures were already established with the help of conventional sterilizing agents.
In the References list there are year descriptions such as “2022a, 2013a, 2022b”. Please follow the guidelines of the Journal.
Author Response
Dear Reviewer,
Thank you for your precious review.
Please you find below our point-by-point responses to your recommendations:
Reviewer 2
The review manuscript entitled “Incorporation of organic additives to enhance in vitro tissue culture for producing genetically stable plants” contains interesting information on summarized data of the application of non-conventional organic supplements in plant cell tissue and organ culture. As such it would be of great interest to the audience. My recommendation is to encourage authors to resubmit after drastic revision of the work following the remarks below.
The main conclusions of the presented review, that non-conventional additives are superior to conventional ones in terms of preservation of genetic stability of in vitro cultured plants lie on the subjective correlation of two groups of non-related datasets.
Thus, Table 2 presents data on a bibliographical survey of a number of examples of genetic variation cases in conventional tissue culture and Table 3 summarizes cases of research of other authors in which non-conventional organic supplements were added to the culture media and no or only minor genetic alterations were established.
In order to make a sound conclusion, comparison has to be made in one and the same experiment. In the case of the present manuscript, please try to change the focus of summary of data in order to keep the information of both Table 2 and Table 3. Or, which I would recommend, keep only a few examples of Table 2 as a text, but strongly expand and improve the manuscript in a different aspect of serious scientific representation of what non-conventional organic supplements are and keep only the data of Table 3.
Table 2 was deleted, only examples were kept as a text.
Conclusion
The concretely reviewed bibliography in the manuscript does not provide examples of solving land degradation and desertification problems. The accent of the work is different and the accent of the conclusion has to be related to the main idea of the work. The issue with land degradation has also been used in supporting the review topic above. Please, change the focus of the work according to its content.
The focus of the work was changed according to its content.
In addition, below are only a few non-exclusive concrete remarks on the scientific writing of the manuscript.
Title
The term “organic additives” standing in the title does not represent concretely what types of organic additives are being reviewed in the work.
The title was changed to ‘Incorporation of organic growth additives to enhance in vitro tissue culture for producing genetically stable plants’
Abstract
In the Abstract and everywhere throughout the text: the expression “true-to-true type” has to be corrected with “true-to-type”.
True-to-true type was replaced by true-to-type across the manuscript.
The whole abstract has to be modified in accordance with the modified content of the re-worked manucript.
The abstract was modified in accordance with the modified content.
“Organic substances”: the same remark as the title. The term has to be substituted with a more concrete and informative one. Conventionally used plant growth regulators, sucrose, vitamins are also organic substances.
Also application of amino acids (for example glycine) is not a non-conventional approach such as fruit juice or coconut water, etc.
The term organic substances was substituted by organic growth additives.
Introduction
Expressions such as “several ecosystems” are not informative and not concrete.
The expression several ecosystems was replaced by agro-silvo-pastoral ecosystems.
Line 67and 70 and elsewhere throughout the text: “synthetic”, “synthetic materials”: The conventionally used tissue culture media ingredients are not necessarily always synthetic.
The word synthetic was deleted.
Chapter 2 “Supplementation of organic additives…..”: the main point of comparison of conventional vs. non-conventional organic additives in tissue culture is not that the first ones are synthetic and the second group - natural. They differ principally in their origin and chemical content. This has to be very well and scientifically-based presented in the manuscript. The reason that the non-conventional organic additives DO WORK in tissue culture is that they contain certain components which serve the SAME functions as of the conventional tissue culture media components (auxins, cytokinins, vitamins, etc). Please expand the review by adding scientifically based discussion on the chemical content and mechanism of action of the non-conventional organic tissue culture additives which are being discussed in the present work.
Till date there are no reports on the mechanism of action of these additives. They are reported to be undefined mixtures of organic nutrients and growth factors (Manawadu et al. [14], Thejaswini et al. [19] Saad and Elshahed [51]). They have certain components whose effect on plant growth and development is unknown. There are also uncharacterized growth substances present in these supplements. Due to this unknown complex nature, their use may lead to poor growth of plantlets or production of unwanted compounds in the media. Hence, it is beneficial to overview the results of experiments with different types and quantities of complex organic supplements for the growth of plant tissues under in vitro conditions.
Everywhere throughout the text: (1) Please, arrange references chronologically when in brackets, (2) References have to be numbered according to their appearance in the text. Please follow the guidelines of the Journal.
References were arranged chronologically and numbered according to their appearance in the text.
Table 1: Please, insert a system to define the order of records in Table 1: are they arranged according to the alphabetical order of species/component/other, or according to the type of organic supplement applied, or the type of tissue culture or the type of effect achieved or other? Does the table follow the numbers of grouping the supplements in 2.1, 2.2, etc?
The table follows the number of grouping the supplements in 2.1, 2.2 and 2.3.
Below tables it is written “Author’s construction”. Table presentation of data should always be author's construction; otherwise reprinting a table from another source would be a case of plagiarism. So, it is not necessary to explicitly mention “Author’s construction”.
“Author’s construction” written below tables was deleted.
The respective Tables in which the discussed information is contained have to be cited throughout the text.
Tables are cited throughout the text.
2.3. Essential oils: Please, specify for the different cases if essential oils were included as components in the medium and the explants - cultivated in their presence? Or were they used as initial sterilizing agents instead of conventional sterilizing agents in tissue culture initiation.
For example in reference [42], the essential oils were included in media after cultures were already established with the help of conventional sterilizing agents.
In references [45-47] the essential oils were included as components in the medium and the explants were cultivated in their presence.
In reference [48], the essential oils were used as initial sterilizing agents instead of conventional sterilizing agents in tissue culture initiation.
In the References list there are year descriptions such as “2022a, 2013a, 2022b”. Please follow the guidelines of the Journal.
Year descriptions were deleted from the references list.
Thank you again for your consideration and service.
Sincerely,

Round 2
Reviewer 2 Report
The authors have considerably improved the manuscript and I suggest it to be published in its current form.